# Development and Initial Validation of the in-Session Patient Affective Reactions Questionnaire (SPARQ) and the Rift In-Session Questionnaire (RISQ)

**DOI:** 10.3390/jcm12155156

**Published:** 2023-08-07

**Authors:** Alberto Stefana, Joshua A. Langfus, Eduard Vieta, Paolo Fusar-Poli, Eric A. Youngstrom

**Affiliations:** 1Department of Brain and Behavioral Sciences, University of Pavia, 27100 Pavia, Italy; paolo.fusar-poli@unipv.it; 2Department of Psychology and Neuroscience, University of North Carolina at Chapel Hill, Chapel Hill, NC 27599, USA; langfus@unc.edu (J.A.L.); eay@unc.edu (E.A.Y.); 3Bipolar and Depressive Disorders Unit, Hospital Clinic, IDIBAPS, CIBERSAM, University of Barcelona, 08028 Barcelona, Catalonia, Spain; evieta@clinic.cat; 4OASIS Service, South London and Maudsley NHS Foundation Trust, London SE5 8AZ, UK; 5Early Psychosis: Interventions and Clinical-Detection (EPIC) Lab, Department of Psychosis Studies, Institute of Psychiatry, Psychology & Neuroscience, King’s College London, London SE5 8AF, UK; 6Helping Give Away Psychological Science (HGAPS), 501c3, Chapel Hill, NC 27599, USA

**Keywords:** therapeutic relationship, affective reaction, emotional reaction, in-session process, self-report measure

## Abstract

This article discusses the development and preliminary validation of a self-report inventory of the patient’s perception of, and affective reaction to, their therapist during a psychotherapy session. First, we wrote a pool of 131 items, reviewed them based on subject matter experts’ review, and then collected validation data from a clinical sample of adult patients in individual therapy (*N* = 701). We used exploratory factor analysis and item response theory graded response models to select items, confirmatory factor analysis (CFA) to test the factor structure, and *k*-fold cross-validation to verify model robustness. Multi-group CFA examined measurement invariance across patients with different diagnoses (unipolar depression, bipolar disorder, and neither of these). Three factors produced short scales retaining the strongest items. The in-Session Patient Affective Reactions Questionnaire (SPARQ) has a two-factor structure, yielding a four-item Negative affect scale and a four-item Positive affect scale. The Relationship In-Session Questionnaire (RISQ) is composed of four items from the third factor with dichotomized responses. Both scales showed excellent psychometric properties and evidence of metric invariance across the three diagnostic groups: unipolar depression, bipolar disorder, and neither of these. The SPARQ and the RISQ scale can be used in clinical or research settings, with particular value for capturing the patient’s perspectives about their therapist and session-level emotional processes.

## 1. Introduction

Emotions are a central pillar of our human existence, and evolved as a biological mechanism for managing critical interpersonal interactions and other crucial life tasks [1]. Consequently, it is only natural that emotions are an integral component of both therapy-specific and nonspecific therapeutic processes [2,3] and play a substantial role in influencing outcomes at the level of individual sessions and the broader course of treatment [4,5,6].

Facilitating the patient to become aware of the emotions experienced during the session and to make constructive use of them is a key path to therapeutic change [7,8]. Among the range of emotions experienced in therapy, those directed toward the therapist hold particular significance, offering a valuable source of clinical information about their personality characteristics [9,10] and psychological functioning in the therapeutic relationship [11,12] capable of informing more effective therapeutic interventions [13,14] and thus improving outcomes [15,16,17,18,19,20].

The challenge lies in allowing the therapists to effectively focus on the patient’s emotions within the therapy session, supporting them in navigating, experiencing, accepting, and ultimately transforming these emotions [21,22,23]. Carefully and systematically assessing how the patient perceives, experiences, and reacts to the therapist during therapy sessions as part of the clinical work can facilitate the understanding of the nature of the patient’s core intrapsychic problems and maladaptive schemas in interpersonal relationships [24,25]. However, the current landscape of therapeutic research and practice reveals a gap.

Despite numerous tools developed to assess emotions and emotional expression [6,26,27], only a few formally integrate attention to emotional processes within the unique dyadic context of the therapeutic relationship. Some work has directly examined emotional reactions and processes during the therapy encounter, using methods such as analysis of transcripts of sessions [28], microanalytic coding of video [29,30], or fundamental frequency of vocal expression [31], as well as long, clinician-reported measures [32,33]. Relatively less work has been undertaken to formally incorporate attention to dyadic affective processes into most psychotherapy modalities.

There currently does not exist a brief assessment tool that could be used to self-monitor affective reactions toward the clinician likely to influence the therapeutic process. This tool should be feasible to use in settings without access to more work on the part of the single therapist or more labor or analytically intensive methodologies used in research-oriented clinical settings. Existing self-report scales for emotion tend to focus on trait affect, or else are state measures that are decontextualized [34,35]. Other scales that measure constructs such as working alliance [36,37] only indirectly include affective content. An ideal tool would provide information about positive affect (e.g., liking, feeling understood, supported, and accepted) and negative affect (including both withdrawal-oriented emotions such as depression, shame, and anxiety, as well as approach/aggressive emotions such as anger and irritation). Additionally, they should be short enough to use repeatedly during a course of treatment. Embedding emotions in the context of in-session events is also likely to help different state emotion from more stable trait effects and reduce mood-congruent biases and halo effects [38].

### Aim

The current study aimed to develop a patient-report measure of patterns of thought, feeling, and behavior activated and experienced in the therapeutic relationship that is clinically sophisticated, psychometrically valid, and easy enough to administer in real-world psychotherapeutic settings for both clinical and research purposes.

## 2. Materials and Methods

### 2.1. Procedures and Sample Characteristics

Eligibility criteria were being 18 years or older, fluent in English, and currently engaged in individual psychotherapy treatment for a diagnosed mental disorder. Participants meeting the study criteria were recruited via two online patient registers (i.e., ResearchMatch and Research for Me) from March through April 2022. ResearchMatch is a disease- and institution-neutral, United States national registry to recruit volunteers for clinical research [39] created by several academic institutions and funded in part by the National Institutes of Health (NIH) National Center for Advancing Translational Sciences (NCATS). Research for Me is a community of volunteers that serves as the central entry point for patients and community members interested in engaging with research at UNC; it was created by the North Carolina Translational & Clinical Sciences Institute (NC TraCS), the integrated hub of the NIH CTSA program at UNC-CH. Evidence indicates that participants recruited through online research platforms are consistent in their self-reported demographic and psychological information, and do not use deception when not financially incentivized [40]. Participants completed an anonymous online survey on Qualtrics, which lasted an average of 15 min.

A small pilot study tested the items on a convenience sample, debugging the Qualtrics programming and checking the wording and clarity of instructions. Participants were then recruited via ResearchMatch and Research for Me. Inclusion criteria were being 18 years or older, fluent in English, and in psychotherapy treatment for a mental disorder; exclusion criteria were deliberately kept minimal. These exclusions were having been declared legally incompetent or having a support administrator.

### 2.2. Item Generation

We followed best practices in scale development, starting with item generation [41,42]. We began with theoretical models motivating item content. Our affective models included the widely accepted positive affect and negative affect model of emotions [43]. We examined the item pool for the extended Positive Affect and Negative Affect Scales (PANAS-X) for candidate content [44]. We enriched it by also considering emotions related to the social/interpersonal dominance dimension [45], which separates emotions such as fear, guilt, and shame (strongly negative valence, but low dominance) from anger, social disgust, and contempt (also strongly negative, but high dominance). We accomplished this by reviewing and including exemplars of discrete emotions that might have distinct motivational properties, including high and low dominance negative emotions [35]. From an evolutionary perspective, these types of emotions served different functions, including fight versus flight in threatening situations [46], and shutting down to conserve resources when helpless [47,48]. We also considered models from the therapeutic process literature, covering affective, cognitive, and behavioral responses, drawing from the clinical-theoretical and empirical literature on transference [9,15,49,50,51] and related concepts [36,52,53].

Item generation combined inductive methods, looking at existing scales for discrete emotions [35], as well as therapy process scales cited above. We also used deductive methods, with experienced clinicians generating items reflecting affective features of good and challenging sessions. The initial item pool was reviewed for content validity by six experienced clinicians, three of whom primarily identify as cognitive-behavioral, two as eclectic, and one as psychodynamic. Six items were eliminated as redundant or poorly worded, and more than a dozen were reworded and reviewed again.

Authorities recommend generating a large initial item pool, much more extensive than the intended scale [41,54]. The longest process scales with which we were familiar suggest an outer limit of around 40–45 simple items for scales intended to be repeated regularly during therapy (e.g., OQ [55]). Our item pool was thus more than triple the size of the longest we would consider feasible for a working instrument. The items were written in everyday language so that the questionnaire could be completed by people with different educational levels. Of note, items were written to sample from positive as well as negative affective domains, ensuring breadth of coverage and avoiding the pitfall of having all items only assessing negative constructs. However, items were written so that each was unipolar. For example, there were separate items for happy and sad affect, rather than one bipolar item ranging from sad to happy, and the response anchors ranged from “Not at all true” to “Very true” for each. These design decisions reflected the current understanding of the measurement of state affect (vs. trait affect or mood) [56], as well as recommendations about reducing cognitive load and improving response accuracy [41]. Items used a 5-point Likert-type scaling, and the instructions directed respondents to think about their most recent therapy session before responding, as our goal was to create a measure of the current process, not attitude towards treatment (as the target audience is people already in therapy), and not yet another measure of temperament or personality. The Flesch–Kincaid readability index was a 7.25, corresponding to a “7th grade reading level—fairly easy to read”.

### 2.3. Additional Measures

The participants completed a 7-item demographic and clinical data form, which recorded their age, biological sex, the clinician’s sex, the frequency of therapy sessions, the length of the course of treatment, and the patient’s clinical diagnoses. Participants were asked to think about their most recent psychotherapy session, read a series of statements that people in psychotherapy might use to describe how they feel toward their therapist (e.g., “During my last therapy session, I felt happy to see my therapist”), and rate each of them on the extent to which each was true of the way they felt during that session. They were asked to respond using a five-point Likert scale: 0 = not at all true, 1 = a little true, 2 = somewhat true, 3 = a lot true, and 4 = very true. Higher scores indicated greater levels of affective reaction.

### 2.4. Statistical Analyses

Scale development followed best practices [44,57,58], and data analyses followed steps similar to prior work [59]. Specifically, the large pool of candidate items was evaluated using descriptive statistics (minimum, maximum, standard deviation, skewness, and kurtosis), checking suitability of individual items for inclusion in subsequent analyses. Those showing insufficient variability were dropped. The Kaiser–Meyer–Olkin test and the Bartlett test of sphericity verified the suitability of the data for factor analysis [60]. Parallel analysis using multiple factor retention methods was run using the R packages *paran* v1.5.2 (Dinno, 2018 [61]) and *EFAtools* v0.4.1 [62] to help determine how many factors might have enough related items to support interpretation. Subsequently, iterative exploratory factor analysis (EFA) using the R package *EFAtools* v0.4.1 [62] analyzed a matrix of the inter-item correlations using polychoric estimation and a PROMAX rotation to achieve simple structure. Items with factor loading less than 0.40 and those with more than 0.30 on two or more factors were removed [63,64], as the large item pool should allow selection of univocal items for major factors. This strategy facilitates unit-weighted scales, which would be easier to use in subsequent validation studies as well as in later clinical applications.

Item response theory (IRT) analyses were implemented in the R package *mirt* v1.36.1 [65] to estimate a Graded Response Model (GRM) for each scale identified by EFA. Item information and coverage were evaluated based on these models. Final item selection chose items with high information across a wide range of theta (θ) levels. IRT methods provide information about whether items are more helpful at low, medium, or high levels of a factor, as well as changes in scale reliability depending on the levels of the factor. In two instances, two items had similar factor loadings and theta levels, so a pool of clinical experts selected one for retention.

The fit of the final factor solution was tested by conducting CFAs using the R package *lavaan* v0.6-11 [66]. Multi-group confirmatory factor analysis examined measurement invariance of the scales across patients with different diagnoses (unipolar depression, bipolar disorder, neither of these). Furthermore, *k*-fold cross-validation using the R package *kfa* v0.2.0 [67] verified the robustness of our final model. We compared results using ML, MLR, and ULS estimators and also examined the statistical power in combination with the expected parameter changes and modification indices to look for model mis-specification [58].

Reliability statistics for the resulting scales were estimated using raw items and 1000 bootstrapped replications [57]. Correlations between questionnaire and patient demographic-clinical features, as well as treatment characteristics, offered preliminary information about the criterion validity of questionnaire scores. Sensitivity analyses re-ran the main models and reliability statistics after trimming the sample to eliminate extreme response times as a filter for online data collection.

## 3. Results

### 3.1. Sample Characteristics

The scale development sample consisted of 701 adults in psychotherapy for a mental health disorder. Most (80%, *n* = 564) were women. The most common age range was 18 to 29 years (40%, *n* = 282), followed by 30 to 39 years (19%, *n* = 131). Each participant had an average of 2.55 (*SD* = 1.53) DSM diagnoses at the diagnostic category level. Many were in psychotherapy over more than 24 months (48%, *n* = 335), typically at a frequency of two to four sessions per month (71%, *n* = 500). More than half of the participants had their most recent session less than one week prior to the study. Table 1 reports sample demographic and clinical characteristics.

### 3.2. Preliminary Analyses

The Kaiser–Meyer–Olkin test (0.96) and the Bartlett test of sphericity (*p* < 0.001) verified the suitability of the data for factor analysis. Two items were deleted because they had polychoric correlations (with smoothing) of >0.90 with other items.

### 3.3. Item Pool Reduction—Iterative Exploratory Factor Analysis

Horn’s [68] parallel analysis using 5000 iterations with simulated *N* = 701 patients and *k* = 129 items identified six factors with eigenvalues greater than one. A six-factor solution was also found when considering eigenvalues higher than those of the 99th percentile of the simulated eigenvalues using 10,000 iterations [69]. Finally, the Hull method [70] and comparison data [71] indicated four factors.

We conducted the first round of EFA extracting six factors, evaluating these for having adequate indicators (at least four items with strong loading) as well as conceptual coherence. Because we had a large initial item pool, we also eliminated items that cross-loaded without a clear dominant loading, to improve the interpretability of scales based on unit-weighted scores. After iterative EFA rounds, 68 items remained in contention, producing a three-dimensional factor structure. Items showed adequate to strong loadings on the respective factor: smallest loadings were 0.58, 0.33, and 0.44, respectively, for factors 1, 2, and 3; while median loadings were 0.74, 0.75, and 0.65, respectively). The items on factor 1 had relatively low endorsement rates compared to the others, creating positive skew at the item level. We considered whether this might be an artifactual “difficulty” factor. After considering the clinical coherence and ramifications of the item content, we opted to dichotomize this subset of items for subsequent IRT analyses and observed score interpretation, such that endorsing any but the lowest option was treated as a concerning, “yes” response. The items on the other two factors all showed acceptable item distributions and satisfied other guidelines for both factor analysis and graded response modeling.

### 3.4. Item Response Theory

Analyses for the Relationship In-Session Questionnaire (RISQ) used dichotomized items; Samejima’s graded response model [72] evaluated the items for the in-Session Patient Affective Reactions Questionnaire (SPARQ) “Positive Affect” and “Negative Affect” factors. Table 2 reports the item discrimination and difficulty parameters of the final scales. Interestingly, the factors had different ranges of theta coverage, despite efforts to select items across a range of levels. The Positive Affect factor had reliability >0.80 from theta of −2.4 to +1.1, indicating that the items were informative and likely to be endorsed even at low levels of the latent variable. In contrast, the Negative Affect factor showed reliability >0.80 at theta ranging from +0.2 to +2.6, and the RISQ factor had reliability >0.80 between theta +0.9 and +3.4, indicating that these items had more information at high levels of the latent variable. Figure 1 shows the item characteristic curves and reliability for the scale scores.

### 3.5. Confirmatory Factor Analysis

*K*-fold cross-validations were performed with *k* = 3 to verify the robustness of our models. The two-factor model, named SPARQ, had the following fit indices: *X*^2^(*df* = 19) = 36.70, CFI = 0.97, TLI = 0.96, RMSEA = 0.06 (90% CI [0.04, 0.08]), and SRMR = 0.05. The one-factor model, named RISQ, had fit indices above a satisfactory range: *X*^2^(*df* = 2) = 8.25, CFI = 0.97, TLI = 0.93, RMSEA = 0.09 (90% CI [0.00, 0.19]), and SRMR = 0.03. A final set of models pooled the samples to provide a final set of parameter estimates (see Figure 2). The two-factor model of the SPARQ provided an excellent fit for the data: *X*^2^(*df* = 19) = 61.48, CFI = 0.98, TLI = 0.97, RMSEA = 0.06 (90% CI [0.04, 0.07]), and SRMR = 0.05. Similarly, a good fit for the data was provided by the one-factor solution of the RISQ: *X*^2^(*df* = 2) = 14.32, CFI = 0.98, TLI = 0.94, RMSEA = 0.09 (90% CI [0.05, 0.14]), and SRMR = 0.03. Statistical power was high for all expected parameter changes and modification indices; none of the tests indicated model mis-specification. Model fit remained good across estimators.

### 3.6. Invariance Testing with Multigroup CFA

To assess measurement invariance across patients with different diagnoses (unipolar depression, bipolar disorder, and neither of these), multigroup CFA models were fit for the SPARQ and the RISQ, respectively. For the SPARQ, a model with no equality constraints across groups showed good model fit (*X*^2^ = 133.28, *df* = 57, *p* < 0.001, CFI = 0.98, TLI = 0.98, RMSEA = 0.08). Equating the loadings, item intercepts, and item thresholds did not significantly harm model fit (Δ*X*^2^ = 53.9, *df* = 56, *p* = 0.55), providing evidence of metric invariance across the three diagnostic groups for the SPARQ.

For the RISQ, the baseline model showed good fit (*X*^2^ = 17.28, *df* = 6, *p* = 0.19, CFI = 0.99, TLI= 0.97, RMSEA = 0.09). As with the SPARQ, equating the item loadings, intercepts, and thresholds did not significantly harm model fit (Δ*X*^2^ = 3.01, *df* = 4, *p*= 0.56), providing evidence of metric invariance across the three diagnostic groups for the RISQ.

### 3.7. Internal Consistency and Score Precision

Table 3 presents the scale descriptive statistics, reliability estimates, and standard errors. The internal consistencies of the final scales were good [54,57]: RISQ (*k* = 4, McDonald’s omega = 0.74, Cronbach’s alpha = 0.75, average inter-item *r* = 0.43), Positive Affect (*k* = 4, omega = 0.86, alpha = 0.86, average inter-item *r* = 0.61), and Negative Affect (*k* = 4, omega = 75, alpha = 0.74, average inter-item *r* = 0.41). The mean scores on the SPARQ “Positive Affect” and “Negative Affect” scales were, respectively, 10.45 (*SD* = 4.16) and 3.03 (*SD* = 3.11). The mean score on the dichotomized RISQ scale was 0.36 (*SD* = 0.87).

For measures of individual precision, we also included the standard error of the measure (*SEm*) and standard error of the difference (*SEd*), along with critical values corresponding to the reliable change index (RCI) propounded by Jacobson and colleagues (e.g., [73]). The 90% value is 1.65 times the *SEd*, and the 95% is 1.96 times the *SEd*. These provide thresholds as being 90% confident that the patient change between the two evaluations was likely to reflect “real” change rather than measurement error. Jacobson stipulated this as a first condition for his two-part nomothetic definition of “clinically significant change”. The second aspect, transitioning past a benchmark based on clinical and or non-clinical reference distributions, is less applicable here: It is not clear what it would mean conceptually to have a “nonclinical reference group” for scores on a scale measuring emotional reactions during therapy sessions. However, it is feasible to define one of the benchmarks based on the clinical distribution. We estimated the 5th percentile for the Positive Affect score, marking where the score would be concerningly low compared to the clinical distribution, and the 95th percentile for the Negative Affect scale, above which the score would be concerningly high based on this nomothetic comparison.

We also included estimates of minimally important difference (MID), using the *d* of 0.5 operational definition [54]. This estimate of MID is more liberal than Jacobson’s RCI-type methods, but it aligns with patient subjective experiences across a broad swathe of constructs and outcome measures [74].

### 3.8. Criterion Validity

The criterion validity of the SPARQ was examined in relation to patient demographic and clinical features, as well as treatment characteristics (see Table 4). All the correlations were small, with none greater than 0.20. However, as expected, the strongest correlations with factors describing negative attitudes were observed with personality disorder. More specifically, small correlations were observed between any personality disorder and the Negative Affect scale of the SPARQ (*r* = 0.17, *p* < 0.001), and between cluster B personality disorder and the RISQ (*r* = 0.20, *p* < 0.001).

### 3.9. Sensitivity Analyses

Sensitivity analyses trimmed the cases to eliminate the fastest and slowest completion times, a standard check for online surveys [40]. Dropping these 36 cases left a sample with *N* = 665. IRT analyses and reliability coefficients and the CFAs all produced results that were identical or changed only at the second decimal place. Model fit was essentially identical: the three-factor model had a robust CFI of 0.995, TLI = 0.994, RMSEA = 0.030 (0.017 to 0.042 90% CI), and SRMR = 0.043, with all factor loadings large and similar to the untrimmed sample. The criterion correlations showed the same substantive results (all available upon request as an *R* notebook). The pattern of response times was highly positively skewed—the fastest completion times were still close to the median, whereas there were some cases with extremely long responses, often an artifact of not clicking past the “Thank you” at the end. The lack of ultra-fast responders is consistent with ResearchMatch being a register of people volunteering to help with research and not expecting compensation.

## 4. Discussion

The goal of this article was to develop and rigorously evaluate a freely available, short self-reported questionnaire assessing the patient’s perceptions and affective reactions to the therapist after a session (see final scales in Appendix A). Although feedback from patients is subject to biases and distortions [75], it can also be a valuable measure of in-session experience. These sorts of affective processes contribute both to positive outcomes [30] and premature dropout (e.g., [31]). Patient-report also could offer a helpful contrasting source of information, counterbalancing potential bias in therapist relationship and process ratings—a considerable source of error in therapists’ ratings of patient emotional experiences and insight, accounting for 30% of the total variance in scores after accounting for perceived emotional intelligence [76]. Gathering the patient’s perspective about their affective reactions would offer therapists more information (“objective data”, from the patient’s point of view), as well as guiding opportunities to disprove negative interpersonal expectations, enhance insight, and reinforce alliance, and ultimately outcome. The literature on routine outcome monitoring in psychotherapy [77,78] indicates that focusing on affective reactions experienced by the patient toward the therapist [79] may be especially effective for those patients who are not doing well in therapy. Furthermore, psychological assessment itself can be a therapeutic intervention when combined with personalized feedback, able to produce positive clinically meaningful effects, especially on treatment processes [80]. The SPARQ represents a further step toward a measurement feedback system that uses valid, reliable, and standardized measures to improve mental health outcomes [81].

We started with an item pool much larger than the intended final length of the scales, aiming to ensure good coverage of the constructs. We reduced the item pool iteratively using a combination of examining item characteristics, clear univocal loadings, adequate indicators for retained dimensions, and clinical cohesiveness of the set [41,82]. Factor analyses converged on a three-factor solution for the SPARQ that was theoretically coherent, clinically meaningful, and had very good internal reliability and consistency. The Positive Affect factor includes items indicating the patient’s perception of being cared for, appreciated, respected, and guided by the therapist. It delineates a secure and comfortable (from the patient’s perspective) experience of the therapeutic relationship, which appears characterized by trust, affective attunement, and positive working alliance. The Negative Affect factor contained items describing feelings of shyness and shame with the therapist, fear of speaking openly, worry of not being helped, and a sense of personal failure due to their need for help from the therapist. An additional factor, the RISQ, has items describing the patient’s tendency to feel disparaged, belittled, rejected, and attacked.

The dimensions of the SPARQ and the RISQ closely reflect the emotional configurations emerging in psychotherapeutic clinical practice [11,13,83,84] and allow therapists and researchers to identify patients’ affective reactions toward their therapist, measure varying levels of them across sessions, and/or assess their relationship to session and treatment outcome. As such, the SPARQ is likely to prove useful for transference work [79], determining ways in which patients interact with their therapists, and increasing the therapist’s understanding of the types and amount of emotional reactions. The identified dimensions likely reflect a mixture of the patient’s own interpersonal dynamics, partially elicited by the therapist and therapeutic setting, and the interaction of patient and therapist in-session attitudes and behaviors.

The medium-size correlations among the SPARQ and the RISQ indicate on the one hand that these are distinct yet related dimensions, and on the other that was possible for a patient to feel cared for by the therapist even when they felt ashamed, afraid to open up with their therapist, or worried that their therapist could not help them, as well as when they are disappointed due to feeling criticized, attacked, or rejected by him/her.

This study included a preliminary investigation of the new scales’ criterion validity by examining the associations between patients’ affects toward their therapist and diagnosis of mental disorders. We found that patients’ in-session affective patterns were not arbitrary but tended to relate to specific diagnoses in clinically meaningful and predictable ways. Consistent with results from previous studies [9,10,85], personality disorders were related to the negative dimensions of the therapeutic relationship. These results suggest that therapists treating a patient with a personality disorder, notably cluster B personality disorders, can expect some occurrence of negative attitudes and behavior against them. By being aware of this situation, the therapist may be able to provide prompt and effective therapeutic intervention, which, among other things, can help decrease premature discontinuation (which is a particularly high risk in patients who have a personality disorder [86]). At the same time, these associations involved only small to moderate amounts of the reliable variance in the scales, indicating that the scales likely measure variations in affective tone in sessions rather than being driven by depression-distortion or personality biases [38]. Put another way, although the scores may be influenced by patient traits, they are not a redundant measure of symptoms.

The different scales that emerged from the analyses have distinct features, and these suggest somewhat varying roles in the context of therapy and treatment research. The Positive Affect scale is the “easiest”, meaning that it would be typical to have high scores after most therapy sessions. It is worth noting that the average was still towards the middle of the possible range of scores (*M* ~60% of the maximum possible), and there were few at the floor or ceiling. This suggests that typical (or good?) sessions may involve some challenging work, and the goal should not be to aim for “perfect” scores (unlike other consumer rating situations, such as Uber, Yelp, or course evaluations at most institutions). When thinking about Jacobson-style normative benchmarks, we debated whether a goal should be to exceed a high bar (e.g., 95% percentile compared to the reference distribution). Having the patient feel substantially less positive about the session and therapist seems more clearly problematic. Scores <4 occurred in only 5% of cases. In contrast, the Negative Affect scale had a lower distribution of scores, with a mean closer to 20% of the maximum possible range. Being closer to the floor, a benchmark pegged to a location in the 5th percentile is impossible (which is quite common in clinical scales in wide use [87]); 24% of cases had an observed score of zero. For the Negative Affect scale, having session ratings above the 95th percentile seems clearly concerning, corresponding to raw scores of 12+.

The RISQ items were rarely endorsed, as evident from item means and the pronounced shift in the region where the items were informative in the IRT analyses. Yet these also showed the most significant correlations with therapy features and patient diagnoses, underscoring their clinical relevance. More work can evaluate whether these are best used in the original Likert format, which would maximize the variance, but have a strongly skewed distribution, versus dichotomizing as “present at all” versus absent, which reduces variability but is frequently performed with clinical items (e.g., [88]). A third approach would be to use it as a checklist where the patient simply checked the box, and endorsing any of them would be a warning flag, given the combination of rarity and severity. Even that most liberal definition only occurred in 25% of cases. Used as a suite, the current data with the three scales suggest that it could be worth checking on the therapy process if Positive Scores fall below 4, Negative Scores rise above 10, and any of the RISQ items are endorsed. These offer provisional operational definitions for investigation in new samples.

The availability of tools such as these that are feasible to use in clinical settings opens up an important set of questions about how best to incorporate these into ongoing treatment. A related consideration would be who is the intended audience for the scores. The focus on the client (trying to improve the therapy process from their own perspective), the therapist, or the clinical supervisor each involves a different interpretive frame and set of goals, and perhaps different ethical considerations.

### Strengths, Limitations, and Future Directions

There were several strengths associated with this study. The development of a self-report questionnaire is a novel contribution to the operationalization of patients’ affective reactions to the therapist, which represents a key component of the therapeutic relationship. The SPARQ has excellent psychometric properties, can be completed in less than three minutes, and is easy to score. Moreover, the development itself followed best practices and utilized a combination of traditional and modern test theories. Finally, a large clinical sample was used.

There were also several limitations that should be addressed, including both technical and conceptual issues. The first is the exclusive reliance on the patient as the sole informant. Patients’ perceptions do greatly matter, but they present only one piece of a complex system. The second limitation concerns the possible bias in patients’ self-reporting their own affective, cognitive, and behavioral reactions. Traditionally, reports on the patient’s emotional responses toward the therapists are obtained via clinicians or external observers/raters. However, the same issues hold true for measuring countertransference (i.e., the therapist’s affective, cognitive, and behavioral responses toward the patient), but there is a body of literature that provides support for using clinicians’ ratings of countertransference [18,89,90,91]. Patient ratings of their own affective responses make particular sense when considering the clinical importance of assessing the subjective emotional experience of the patient. Third, the psychometrics of the 15 extracted items should be confirmed in a sample where they are not embedded in the larger original item pool, checking that performance is similar without context effects. These tend to be small with scales that are homogeneous and have strong factor loadings, as is the case here, but they remain worth corroborating. Although we used *k*-fold cross validation and a large patient sample, our CFA models were still based on the same clinical sample as the exploratory analyses. It will be important to replicate with patients ascertained in a different way or for additional clinical issues to address generalizability. Fourth, now that a reduced item set has been identified, systematic exploration of its dependability, retest stability, and sensitivity to treatment effects will be an important next step in validation [57].

Future research using the SPARQ should examine affective states and processes from multiple perspectives to assess its validity and correlates and understand how self-reported affective reaction relates to therapists’ perceptions of this phenomenon. Future studies should also investigate how this measure relates to the process and outcome of therapy, as well as to other components of the therapy relationship, especially countertransference, working alliance, and real relationships. Finally, longitudinal research will add to our how these processes unfold over the course of psychotherapy and predict different trajectories. In sum, a major next step would be to examine multivariate models for combining information from different therapy processes and outcome measures, different informants (e.g., therapist and external observer), patient’s personal family history/demographic/clinical characteristics, and therapist’s personal characteristics and in-session attitudes and behaviors (including the therapeutic interventions) to examine incremental validity and to develop decision support algorithms and optimal sequences.

## 5. Conclusions

The patient’s experience and perceptions of their psychotherapist must be accurately identified (“diagnosed”) and discussed with the patient in a form and at a time that suits them. This article details the development and validation of two new brief self-report measures of the patient’s affective reactions toward their psychotherapist. Both the SPARQ and the RISQ show excellent psychometric properties and are short and easy for patients to complete on their own. The results support the potential usefulness of these scales in assessing the patient’s affective responses during therapy, and they provide initial evidence that these measures are appropriate for research and clinical use in individual psychotherapy settings. By enabling patients to rate their own affective reactions toward their therapist on a carefully developed, normed questionnaire, we turn patients’ emotional experiences into quantifiable dimensions that can be analyzed, used to guide clinical interventions, and employed as indices of clinical change. These questionnaires may also be a useful tool in clinical supervision for psychotherapy trainees.

## Figures and Tables

**Figure 1 jcm-12-05156-f001:**
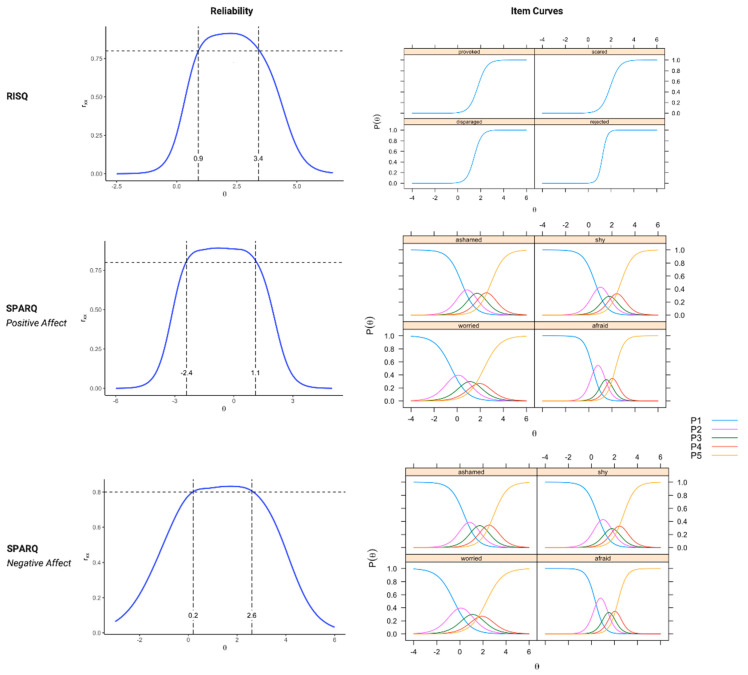
Item Option Characteristic Curves and Reliability for the Scale Scores. The curves on the right show the threshold where a patient’s probability changes from a lower to the next higher option on the item. The reliability curves transform test information into a reliability estimate (between 0 and 1.0) and show how reliability changes at low (negative θ values), average (θ = 0), and high levels (positive θ) of the underlying factor.

**Figure 2 jcm-12-05156-f002:**
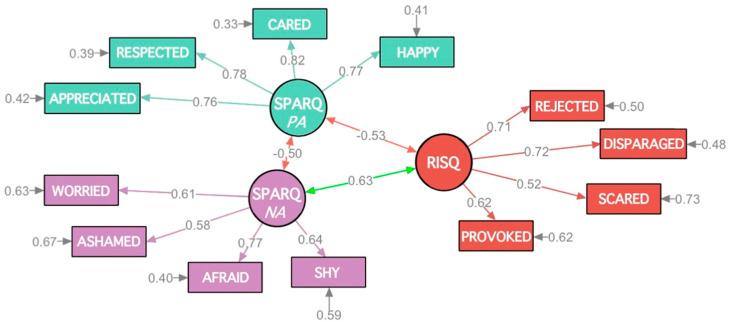
Measurement Model from Confirmatory Factor Analysis (*N* = 701) presenting a fully standardized solution using robust maximum likelihood estimation. Note: This Figure presents abbreviated item content for both the RISQ and the SPARQ items.

**Table 1 jcm-12-05156-t001:** Demographics, Clinical, and Treatment Characteristics of Participating Patients (*N* = 701).

Demographics	% (*n*)
Biological sex	
Male	18% (126)
Female	80% (564)
I prefer not to say	2% (11)
Age (years)	
18–29	40% (282)
30–39	19% (131)
40–49	15% (104)
50–59	15% (105)
≥60	11% (79)
**Clinical Characteristics**	
Average number of diagnoses, *M (SD)*	2.55 (1.53)
Any anxiety disorder	75% (529)
Any (unipolar) depressive disorder	54% (378)
Any bipolar or related disorder	17% (117)
Any personality disorder	13% (93)
Any trauma- and stressor-related disorders	30% (209)
**Treatment Characteristics**	
In psychotherapy from	
0 to 3 months	18% (124)
4 to 6 months	10% (72)
7 to 12 months	12% (86)
13 to 24 months	12% (84)
>24 months	48% (335)
Session frequency	
≤1 per month	24% (171)
2 to 3 per month	35% (244)
1 per week	37% (256)
≥2 per week	4% (30)
Therapist’s biological sex Female	77% (539)
Patient–Therapist biological sex match Same-sex	74% (521)

* Note: *N* sums to more than 701 because cases could have more than one diagnosis.

**Table 2 jcm-12-05156-t002:** Item Option Characteristics for the three factors based on IRT models.

	Item Content	α	β1	β2	β3	β4
**Factor 1**	I felt disparaged or belittled by my therapist	3.05	1.46			
RISQ	I felt rejected by my therapist	4.73	1.28			
	I felt provoked or attacked by my therapist	2.31	1.84			
	I felt scared, uneasy, like my therapist might harm me	2.23	2.06			
**Factor 2**	I felt respected by my therapist	2.60	−2.07	−1.32	−0.73	0.19
SPARQ Positive	I felt my therapist cared about me	2.98	−1.95	−1.04	−0.40	0.51
	I felt happy to see my therapist	2.47	−1.83	−0.91	−0.25	0.61
Affect	I felt appreciated by my therapist	2.44	−1.29	−0.57	0.22	1.14
**Factor 3**	I felt worried my therapist couldn’t help me	1.42	−0.51	0.68	1.54	2.32
SPARQ Negative Affect	I felt afraid to spoke my mind, for fear of being judged, criticized, disliked by my therapist	2.67	0.34	1.26	1.77	2.32
	I felt ashamed with my therapist about my fantasy, desires, mindset, behavior, or symptoms	1.73	0.37	1.32	2.12	2.95
	I felt shy, like I wanted to hide from my therapist or end the session early	1.96	0.54	1.48	2.09	2.79

**Table 3 jcm-12-05156-t003:** Descriptive statistics, internal consistency reliability, precision, and inter-scale correlations.

	RISQ	SPARQ	
		Positive Affect	Negative Affect
**Descriptive statistics**			
Potential Range	0 to 4	0 to 16	0 to 16
Observed Range	0 to 4	0 to 16	0 to 16
Mean, *SD*	0.36 (0.87)	10.45 (4.16)	3.03 (3.11)
POMP, *SD*	9.00 (21.75)	65.31 (26.00)	18.94 (19.44)
Skew	2.67	−0.58	1.33
Kurtosis	6.75	−0.53	1.57
Standard Error of Measurement (SE_m_)	0.44	1.56	1.55
Standard Error of Difference (SE_d_)	0.63	2.20	2.50
**Internal consistency reliability**			
*X*^2^/*df*	7.16	3.24	
CFI	0.98	0.98	
TLI	0.94	0.97	
RMSEA	0.09	0.06	
SRMR	0.03	0.05	
Average inter-item *r*	0.43	0.61	0.41
Alpha	0.75	0.86	0.74
Omega total	0.74	0.86	0.75
**Clinical change benchmarks**			
90% Critical Change	1.02	3.63	3.63
95% Critical Change	1.21	4.31	4.61
Minimally important difference	0.44	2.08	1.55
Jacobson benchmark threshold (5% tail)	--	LB: 2.30	UB: 9.13
**Scale correlations**			
SPARQ—Positive Affect	−0.45 *	1	
SPARQ—Negative Affect	0.49 *	−0.40 *	1

Note: SPARQ = in-Session Patient Affective Reactions Questionnaire; LB = Lower Bound; POMP = Percentage of Maximum Possible; RISQ = Relationship In-Session Questionnaire; UB = Upper Bound. RISQ uses dichotomized answers as “Not at all true” versus the other options. Minimally important difference was operationally defined as *d* = 0.5. * *p* < 0.0005, two-tailed.

**Table 4 jcm-12-05156-t004:** Criterion Validity Correlations with Patient Diagnoses, Demographics, and Objective Therapy Characteristics.

Criterion Variable	RISQ	SPARQ	
		Positive Affect	Negative Affect
Age	−0.04	0.05	−0.15 ***
Sex	−0.06	0.04	−0.06
Average # of diagnoses	0.04	−0.07	0.17 ***
Any anxiety disorder	−0.10 *	−0.00	0.03
Any bipolar disorder	0.05	−0.05	0.03
Any depressive disorder	−0.14 ***	0.06	0.02
Any personality disorder	0.20 ***	−0.11 *	0.17 ***
Cluster A PD	0.08 *	−0.08	0.08 *
Cluster B PD	0.19 ***	−0.12 **	0.13 **
Cluster C PD	0.07	−0.08	0.09 *
Any trauma- and stressor-related disorder	−0.02	0.04	0.05
Therapy length	−0.11 *	0.13 **	−0.05
Session frequency	0.03	0.10 *	0.06
Therapist’s sex	−0.13 **	0.10 *	−0.07
Patient–Therapist sex match	−0.10 *	0.09 *	−0.12 **

Note: Coefficients are point-biserial correlations for dichotomized variables, point-biserial correlations for dummy-coded categorical variables, Spearman correlations for ordinal variables, and Pearson correlations for continuous variables. * *p* < 0.05, ** *p* < 0.01, *** *p* < 0.001.

## Data Availability

Both the data and the analysis code that support the findings of this study are available from the corresponding author upon reasonable request.

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
