# Peer review of "Development and Initial Validation of the in-Session Patient Affective Reactions Questionnaire (SPARQ) and the Rift In-Session Questionnaire (RISQ)"

_jcm, 2023, doi:10.3390/jcm12155156_

Round 1

Reviewer 1 Report

Dear authors, 

The paper discusses and develop a very important tool. I JUST  have two comments:

1-     Abstract: This refer to?

2-      Introduction: “There currently does not exist a brief assessment tool that could be used to

self-monitor affective reactions toward the clinician likely to influence the therapeutic

process that would be feasible to use in settings without access to more

work on the part of the single therapist or more labor or analytically intensive

methodologies used in research-oriented clinical settings” too long sentence makes the reader unable to follow the context in meaningful way

Long sentences need to be short for being easily understandable, 

Author Response

Thank you for your positive evaluation.

We fixed the error in the abstract ("This" referred to the article) and split the long sentence.

Reviewer 2 Report

Title: The title "Development and Initial Validation of the in-Session Patient Affective Reactions Questionnaire (SPARQ) and the Rift In-Session Questionnaire (RISQ)" is self-explanatory and effectively conveys the research objectives.

Abstract: The abstract is concise and effectively summarizes all the relevant information and conclusions of the study.

Introduction: The authors have provided a comprehensive account of the research question and aims, making the concept of the study clear to the readers.

Materials and Methods: The item generation process is well explained, detailing the theoretical models used and the large initial item pool. The use of iterative exploratory factor analysis (EFA) with item response theory (IRT) analyses is well-justified. The fit of the final factor solution was appropriately tested by conducting confirmatory factor analyses (CFAs).

It would be beneficial to move the description of Table 1, which includes demographics, clinical, and treatment characteristics of the participating patients (N=701), from the Materials and Methods section to the Results section. Doing so would enhance the clarity of both sections.

Results: The results are appropriately presented in both tables and text, ensuring a comprehensive understanding of the findings.

Discussion: The study provides a preliminary investigation of the new scales' criterion validity by examining the associations between patients' affective reactions towards their therapist and their mental disorder diagnosis. Furthermore, the results are compared with findings from previous studies, enhancing the significance of the study.

Overall, the article is well-written and provides valuable insights into the development and validation of the in-session patient affective reactions questionnaire and the rift in-session questionnaire. The study's methodology and statistical analyses are robust, and the authors have acknowledged the study's limitations thoughtfully. This research contributes significantly to the field and will be of interest to both researchers and practitioners.

However, some minor improvements can be made by moving the demographics, clinical, and treatment characteristics table to the Results section for enhanced clarity. Additionally, any ethical considerations and implications of the study could also be briefly discussed.

Author Response

Thank you for your positive evaluation.

We moved Table 1 and added a paragraph on ethical considerations and implications of the study in the Discussion section.